

# A semi-continuous study on the toxicity of atmospheric particles using a versatile aerosol concentration enrichment system (VACES): development and field characterization

Xiaona Shang[1], Ling Li[1, 2], Xinlian Zhang[1], Huihui Kang[1], Guodong Sui[1], Gehui Wang[2], Xingnan Ye[1],
Hang Xiao[3], Jianmin Chen[1, 2, 3*]

[1] Shanghai Key Laboratory of Atmospheric Particle Pollution and Prevention (LAP[3]), Department of Environmental Science & Engineering, Institute of Atmospheric Sciences, Fudan University, Shanghai 200438, China

[2] Institute of Eco-Chongming, 3663 N. Zhongshan Rd., Shanghai 200062, China

[3] Institute of Eco-Environment, University of Chinese Academy of Sciences, Beijing 100080, China

*Correspondence to: Jianmin Chen (jmchen@fudan.edu.cn)

**Abstract:** The toxicity of atmospheric particles directly associates to health effects, but its online monitoring has not yet been implemented due to low-concentration toxic components and high measurement detection limit. To solve the detection problem, we extended a versatile aerosol concentration enrichment system (VACES) for toxicity aerosol measurement and firstly used VACES to provide a comparison of toxicity between non-concentrated and concentrated aerosols in ambient air. Through

optimizing the technical parameters, the total concentration (number or mass), the concentration of chemical components, and the toxicity were all increased by approximately 7 to 10 times in VACES. In particular, ambient aerosols with toxicity below the detection limit, were detected significantly after concentration. Moreover, comparable enrichment factors and similar trends before and after enrichment in toxicity were observed over time, suggesting that the toxic properties of ambient aerosols do not change in VACES. Whereas, changes in $PM_{2.5}$ concentrations and their toxicity do not always correlate well, which was

probably driven by the combined toxicity effect of chemical components. Thereby, it hints a necessity to further study on the toxicity effects of chemical compositions. All above also imply that VACES could provide technical support for the future measurement of online atmospheric particulate toxicity, which is currently performed at high-concentration environment, long sampling duration, and offline.

## 1 Introduction

Numerous epidemiological studies show strong correlations between atmospheric particulate matter and adverse health effects such as allergy, respiratory diseases, and cardiovascular (e.g. Nel, 2005; Brook et al., 2010; D'Amato et al., 2013; Tapia et al., 2019). The hazardous substances refer to not only the particle masses, but also their chemical components. Li et al. (2019) figured out that particulate matter concentration does not fully reflect the mortality rate in different countries and regions, and emphasized that the formulation of policies should be based on the premise of identifying the most toxic species of particles

in each region.

The measurement of atmospheric particle toxicity directly associates to the assessment of health effects. Currently, most toxicity studies focus on discovering the relationship between particulate matter and the morbidity or mortality of organisms (e.g. Vincent et al., 2001; Cox et al., 2016; Miri et al., 2018), or exploring toxic mechanisms by exposure experiments (e.g. Magnani et al., 2016; Huang et al., 2017; Rychlik et al., 2019). However, the measurement toxicity data are rarely available

because of technical limitations. For instance, it requires a long detection time due to the animal and plant reproduction or cell cultivation (National Research Council, 2006), but the concentration and chemical composition of particulate matter in the atmosphere continue to change over time, especially during severe pollution (Shang et al., 2018a, b). Thereby, a short analyzing time is quite important.

To solve this problem, photobacteria (e.g. Photobacterium phosphoreum) are utilized in the toxicity study of atmospheric

particles, because the detection can be completed in a short time (e.g. within 15 minutes; Hoover et al., 2005) and the cultivation period is only 5 minutes (Jing et al., 2019). However, the detection limit of toxicity using bacteria is much higher than using other organisms such as cells. For example, in Jing's research, samples with a light inhibitory rate of less than 20 % were considered to be non-toxic due to the impact of normal bacteria fluctuations. Whereas, the concentration of atmospheric aerosols is usually far lower than that required for toxic tests, particularly for such a high detection limit, which means a longer

sampling time is required. Nevertheless, long-time sampling may lead to a large loss of volatile substances or chemical reactions in the particles, subsequently resulting in large errors in toxicity analysis (Weiden et al., 2009).

In this respect, aerosol enrichment techniques have been developed and applied to increase aerosol concentrations to meet toxicity detection limits. Among them, the versatile aerosol concentrator enrichment system (VACES) originally developed by

Sioutas et al. (1999) is effectively used to concentrate ambient particles. Since then, it has been widely used for laboratory and

field measurements of particulate matter (De Vizcaya-Ruiz et al., 2006; Steenhof et al., 2011; Plummer et al., 2012; Loxham

et al., 2013), because the physical and chemical properties do not change after concentration (Kim et al., 2001a, b; Wang et al.,

2013). It has also been extended to combine various chemical and physical analysis of particulate matter (e.g. gases, water-

soluble ions, heavy metals, polycyclic aromatic hydrocarbons, cloud condensation nuclei, etc.) (Jung et al., 2010; Freney et

al., 2006; Pakbin et al., 2011; Zhao et al., 2005; Dameto et al., 2019). In addition, VACES has been applied to determine the

relationship between particulate matter and health effects based on exposure experiments (Klocke et al., 2017; Ljubimova et

al., 2018). Nevertheless, although VACES was originally developed to provide technical support for toxicity detection, there

is no direct measurement data to show the change in toxicity between ambient particles and VACES particles.

Therefore, according to the previous design, by optimizing technical parameters, we modified and further developed VACES

to integrate into the toxicity measurements, verified the enrichment effect on physio-chemical concentration and toxicity in

laboratory and field studies, and also investigated the relationship between toxicity and particulate masses.

## 2 Methodology

### 2.1 Design of VACES

VACES uses a saturation and condensation system to rapidly grow particles into super-micron droplets, which are then

concentrated by a virtual impactor. Detailed description of the design of VACES is available in previous studies (e.g. Kim et

al., 2001a, b). Briefly, a heat tube is set inside of a water tank filling with deionized water to provide super-saturation

environment. A chiller (Bilon, China) filled with ethanol (80 %, Hushi, China) cools through the tube to condense saturated

aerosols. The condensed aerosols are drawn into a virtual impactor, where particle concentration in sizes is concentrated to a

desired level by changing the ratio of the minor-to-major air flow controlled by a mass flow controller (MFC, D08-4F,

Sevenstar, China). The concentrated aerosols are dried to their original sizes before being introduced into subsequent analyzing

instruments using a Nafion tubing (MD-700, Perma Pure, USA).



## 2.2 Sampling

The dry air sample was introduced into a differential mobility analyzer (DMA, Model 3081, TSI, USA) coupled to a condensed particle counter (CPC, Model 3775, TSI, USA) for VACES performance testing on the 6th floor of the Department of Environmental Science and Engineering, Fudan University, Shanghai. Open sampling (without cutting size) switched between

ambient air and VACES air every minute at a flow rate of 1.5 L min$^{-1}$ to compare the particle number and mass concentration on the particle size.

Based on performance testing, set the optimal technical parameters to characterize VACES. This study used the enrichment factor (EF), defining as the ratio of the concentrated to non-concentrated particulate concentration, and the enrichment efficiency (EE), defining as the ratio of the measured to theoretical concentration of concentrated aerosol, to examine the

enrichment effect of VACES. The initially determined EF was 10 corresponding to the EE of 100%. For the characterization, on the one hand, polystyrene latex (PSL, size range: 0.2–0.7 µm; Thermo Fisher Scientific, USA) aerosols are atomized in laboratory using an atomizer (Model 9302, TSI, USA) to generate particles of four sizes (200 nm, 300 nm, 500 nm, and 700 nm). The size of the generated aerosols was screened by DMA at the corresponding voltage, and then CPC was used to count the number concentration of PSL aerosol with and without enrichment, respectively, at a flow rate of 0.3 L min$^{-1}$ (Fig. 1).

Similarly, in field study the ambient air instead of the PSL aerosol was directly sucked into VACES. Note that the impact of particle formation in VACES was investigated in this study by removing ambient aerosols using an aerosol filter (ETA Filters, USA), with a maximum flow rate of 50 L min$^{-1}$, forward of the saturator and comparing the number and mass concentrations with and without enrichment in VACES.

After performance testing and VACES characterization, toxicity experiments were performed. The concentrated aerosol was

collected offline at a minor flow rate of 5 L min$^{-1}$ for 30 minutes and one hour, respectively, in a biosampler (SKC, USA) containing 5 ml DI. It is an optimal airflow rate for obtaining the highest collection efficiency when connected to VACES (Kim et al., 2001a). For contrast, the same biosampler was used to collect ambient aerosols in 20 ml DI, but the recommending air flow was 12.5 L min$^{-1}$ regarding to the non-supersaturated aerosols. We also performed hourly VACES online sampling and used a method similar to offline sampling. But one of the differences was that we added a peristaltic pump (BT100-4, HUXI,

China) forward of the biosampler to pump in DI, and then connected the outlet of biosampler to another peristaltic pump to

evacuate the sample into an automatic fraction collector (BS-40A, HUXI, China). Pumping in and out were performed at a scheduled time (59-minute sleep mode and 1-minute work mode) and volume (5 mL). Another change was the particle size of the inlet. We used a sampler (PM-100, WUHAN TIANHONG, China) to cut the particle size to 2.5 μm at a flow rate of 100 L min$^{-1}$ for continuous sampling. Meanwhile, we used a cyclone (Met one Instruments, USA) to sample ambient PM$_{2.5}$ on a 47

mm Teflon filter (Whatman, USA) for 8 hours each and flow at the same 5 L min$^{-1}$ with that of VACES sampling. Then, we obtained 20 sets of non-continuous samples (10 sets of 1-hour samples and 10 sets of 30-minute samples, respectively) and 88 VACES continuous samples with 11 ambient filter samples for toxicity measurement, and compared their toxicity before and after concentration. Note that the filter samples were extracted in 10 ml deionized water via 20-minute sonication and 45 °C heating condition (the same setting with VACES sampling).

**2.3 Measurements**

All samples were filtered using 0.22 μm pore size filters (Collins, China) and 10ml sterile syringes (KDL, China). Thereafter, toxicity assay and water-soluble ion measurement were conducted immediately. Regarding to the toxicity assay, Jing et al. (2019) provided detailed information. In brief, 100 μL of the prepared bacterial suspension was pipetted into cuvettes to measure the luminous intensity as the baseline. Then, the initial luminous intensity was recorded after adding 100 μL of sample.

In 15 minutes, the luminous intensity was recorded again. After subtracting blank intensity tested using NaCl solution (3 %), the light inhibition rate of particles with and without enrichment was calculated, respectively, according to the international standard (ISO 11348-1: 2007) procedures (Water quality, 2007). All samples were tested in triplicate and averaged in present study. To ensure the enrichment effect of VACES system, we also detected water-soluble ions of both ambient and VACES samples collected during continuous sampling period using an ion chromatography (940 Professional IC Vario, Metrohm,

Swiss) integrated with an autosampler (863 Compact Autosampler, Metrohm, Swiss). Moreover, the atmospheric PM$_{2.5}$ concentration was monitored in a nearby state-controlled site (Liangcheng, Hongkou).

## 3 Results and discussion

### 3.1 Performance test of VACES

The performance test of VACES was operated directly in ambient rather than using generated aerosols. Optimization of VACES

is to achieve 10-fold enrichment of ambient aerosol concentration mainly through modulating temperatures of saturator and

chiller, the major air flow, the minor air flow and their flow ratio. By switching the two air pathways (ambient air and ambient

air passing through VACES, respectively) and comparing their number and mass concentrations observed in scanning mobility

particle sizer (SMPS: DMA+CPC), we established technique standards for the desired EF (10) and EE (100 %). Results showed

that the EF of 10 could be achieved for particles larger than ~ 30 nm as setting the optimal parameters of –19 ± 1 °C

condensation temperature, 45 ± 2 °C saturator temperature, 50 ± 1 L min$^{-1}$ major air flow, and 1/10 minor-to-major air flow

ratio, respectively. The corresponding EE ranged from 75 % to 100 % in different size ranges as listed in Table 1. The highest

EE was obtained in size range of 30–100 nm, very close to 100%.

### 3.2 VACES characterization in laboratory and field study

For laboratory VACES characterization, PSL was used to generate four sizes (200, 300, 500, and 700 nm) of particles from an

atomizer, and the size was cut by DMA, and then by CPC the number concentration of VACES particles with and without

enrichment was alternatively measured six times in parallel. The EF calibration line was plotted against the particle number

concentration in four sizes with and without enrichment. It showed a quite high correlation coefficient ($r^2$ = 0.9999) and the

EF of VACES was 10 approximately (Fig. 2).

The characterization of VACES was also investigated in the ambient. Similarly, we measured the number and mass

concentrations of particulate matter in ambient air that was and was not concentrated, respectively. When the concentration

coordinate value of VACES is set to 10 times that of ambient, the two curves almost completely coincide for particle size

greater than 25 nm (Fig. 3a and 3b), indicating that the EE is close to 100 %. In addition, in order to eliminate the influence of

particle formation on the enrichment effect in VACES, an aerosol filter was set forward of the inlet to remove ambient particles.

The particle number and mass size distribution with and without aerosol filters were examined and compared (Fig. 3c and 3d).

It found that the mass peak in VACES only comprised ~ 1 % in total mass concentration and could be neglected. Moreover,





the mass peak always appears with the same particle size and the similar concentration, which indicates that it is most likely a water vapor peak and not a newly formed particle in the system.

**3.3 Toxicity variation of concentrated aerosols in VACES**

The discontinuous samplings were operated on relatively clean and polluted days in the period of Oct. 23 ~ Dec. 11, 2019. The

10 sets of ambient samples with and without enrichment were collected hourly and half-hour using a biosampler, which were sampled at a $PM_{2.5}$ concentration varied from 21 μg m$^{-3}$ to 187 μg m$^{-3}$. During the sampling period, none ambient samples were toxic (light inhibition rate > 20 %, detection limit), and most were far lower than the detection limit (Fig. 4). Whereas, the toxicity of almost VACES samples could be detected evidently. Note that below detection limit (20 %) it is considered to be the normal fluctuation range of the luminescence bacterium, hence the sample has a large uncertainty regarding to toxicity

within this range. It implies that the increase in toxicity caused by concentration enrichment can avoid the inaccurate assessment of particulate toxicity caused within the high-uncertainty range. Overall, the toxicity of concentrated aerosols was elevated by a factor of 8 on average, which was at a similar level with the change of the number and mass concentrations of particles as mentioned above. Even though the EFs varied approximately between 6 and 10 times, some extremes were also observed in both one hour and half an hour sampling. For instance, EF of the sample 5 and 6 in hourly sampling exceeded 10

and EF of the sample 1 in 30-minute sampling was even lower than one, all of which were probably derived by the far low toxicity of ambient samples (light inhibitory is 1.5 %–6.3 %). In particular, the variations of the light inhibition rate between ambient and VACES presented a similar pattern although the EFs were not always same, indicating that the VACES does not change the toxic properties during concentrating process. The EF changes of toxicity were probably attributed to the variation of chemical composition enrichment. In addition, the atmospheric $PM_{2.5}$ concentration could not fully explain the light

inhibitory of ambient and VACES particles (their correlation coefficients were only 0.2–0.5). The weak correlation suggested that the change in the concentration and proportion of toxic chemical components, or even the combined effect of the toxicity of each composition might affect the toxic variation (Akhtar et al., 2014). In this regard, the effects of key toxic components on toxicity changes need to be further studied using VACES.

The continuous sampling was carried out from Dec. 18, 2019 to Dec. 31, 2019, during which the $PM_{2.5}$ concentration varied



from 14 µg m$^{-3}$ to 107 µg m$^{-3}$. In the whole sampling period, the toxicity (light inhibitory) of both ambient and VACES particles exhibited similar trends with the change of PM$_{2.5}$ concentration (correlation coefficients >0.7) (Fig. 5). Meanwhile, the toxicity EFs of PM$_{2.5}$ in VACES reached 7–10 apart from two outliers (Fig. 6). The two extremes of light inhibitory were resulted by the super low values of ambient particles (light inhibitory < 2 %) and they were considered to possess a significantly high uncertainty (Fig. 5). Furthermore, the EFs were also calculated by the concentration of chemical compositions of ambient and

VACES particles in order to ensure the enrichment effect of VACES in real time. Results showed that most samples and chemical compositions were concentrated well around 7.5–9.1 times on average. Regarding to the low EFs of Cl$^-$ in some PM$_{2.5}$ samples, it was likely due to the evaporation during toxicity measurement or the loss in virtual impact of VACES system.

## 4 Conclusions and implications

To achieve detection limits for atmospheric particulate toxicity, a versatile aerosol concentration enrichment system (VACES)

was extended to be integrated with toxicity measurement. The VACES was developed to concentrate atmospheric aerosols by adjusting system parameters and a series of technical standards were established including chiller temperature (–19 ± 1 °C), concentrator temperature (45 ± 2 °C), major air flow (50 ± 1 L min$^{-1}$), and minor-to-major flow ratio (1/10). When set to the optimal parameters, the number and mass concentrations of either the generated PSL particles or ambient aerosols were well enriched by a factor close to 10. Based on the enrichment of VACES, the toxicity of concentrated ambient aerosols was detected

using discontinuous hourly and half-hour sampling and continuous hourly sampling, and their light inhibitory effect on luminescent bacteria was compared with that of non-concentrated aerosols. As atmospheric PM$_{2.5}$ concentration varied from 14 µg m$^{-3}$ to 187 µg m$^{-3}$, the light inhibitory of ambient aerosols were all below detection limit (20 %). After enrichment, however, the light inhibitory increased to a level that was significantly detected for almost samples. Particularly, the enrichment factors (VACES/Ambient) of the light inhibitory and even the chemical compositions of PM$_{2.5}$ changed within a small range

(around 7–10) and was comparable to that for number or mass concentration (around 7.5–10). In addition, toxicity changes were not always consistent with those of particle masses, which might be caused by the combined toxic effects of chemical compositions of particulate matter (Akhtar et al., 2014). Therefore, it provides hints for further studies on the effects of key chemical components on toxic changes in atmosphere. Moreover, the rapid recovery of photobacterium (< 5 minutes), the short

toxicity detection time (15 minutes), and the on-demand increase in toxicity but unchanged toxicity property in VACES system,

all provide technical support for future online toxicity measurement of atmospheric particle.

**Data availability:** Data is available by contacting the corresponding author.

**Competing interests:** The authors declare that they have no conflict of interest.

**Acknowledgements**

This work was funded by National Natural Science Foundation of China (Nos. 21527814, 91843301, 91743202).

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



**Table 1** Enrichment efficiency of ambient aerosols in VACES at different size ranges.

| Particle size | Condensation (–19 ± 1 °C) | Saturation (–45 ± 2 °C) | Major flow (50 ± 1 L min⁻¹) | Minor-to-Major flow ratio (1/10) | Total |
|---|---|---|---|---|---|
| 30–50 nm | 99 % ± 22 % | 85 % ± 10 % | 97 % ± 13 % | 97 % ± 13 % | 98 % ± 8 % |
| 50–100 nm | 100 % ± 12 % | 85 % ± 5 % | 99 % ± 9 % | 99 % ± 9 % | 91 % ± 6 % |
| 100–200 nm | 85 % ± 11 % | 82 % ± 3 % | 78 % ± 12 % | 80 % ± 12 % | 79 % ± 3 % |
| 200–1000 nm | NA | NA | NA | NA | 75 % ± 10 % |

*NA: not available





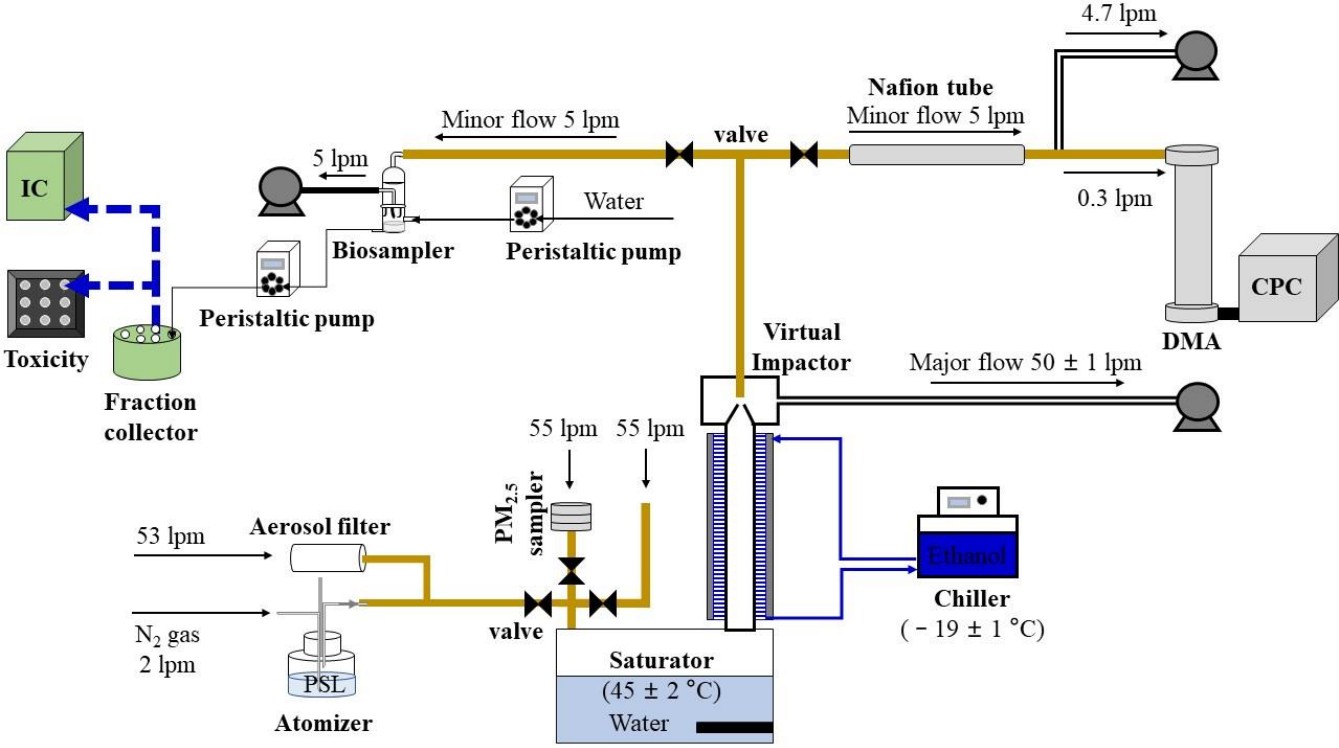

**Figure 1** Set-up for VACES characterization, toxic and chemical experiment.





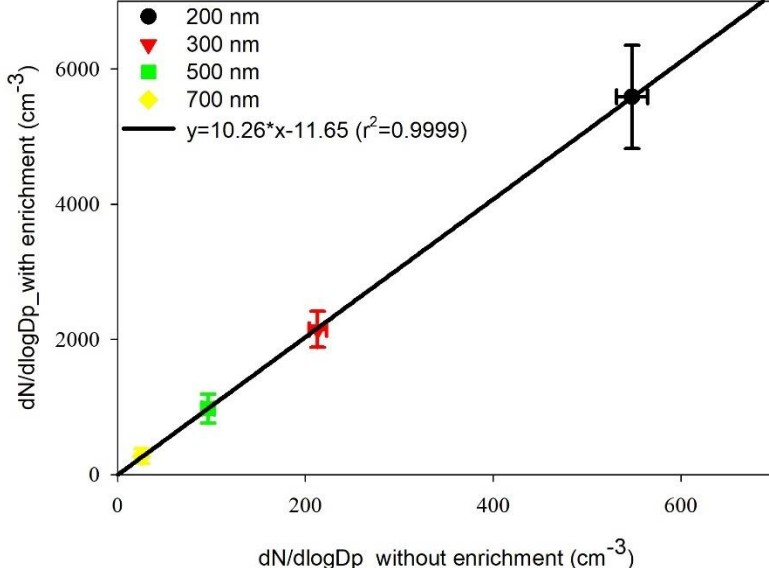

**Figure 2** Calibration of enrichment factor of VACES system using Polystyrene Latex (PSL) aerosol reagent at the size of 200

~ 700 nm, where error bars are standard deviation of six parallel measurements.



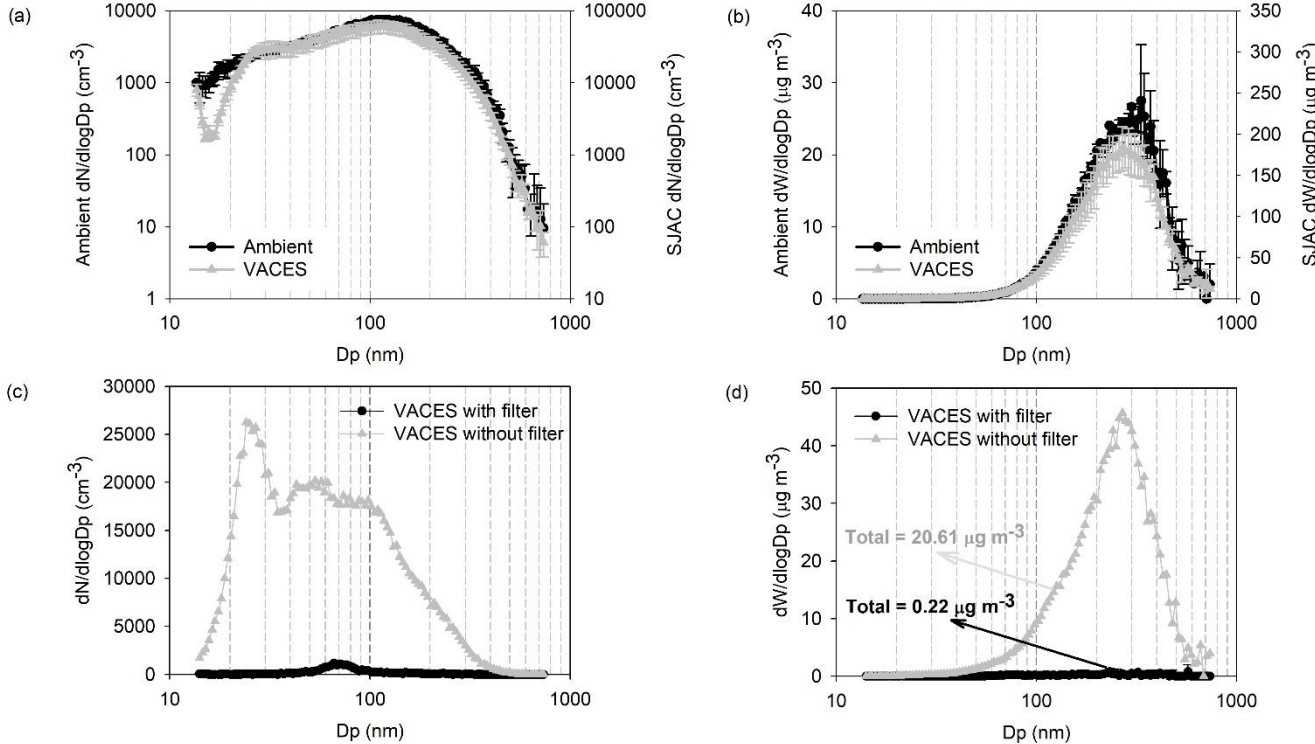

**Figure 3** Particle (a and c) number and (b and d) mass size distribution in ambient and VACES system, error bar is standard deviation of three parallel experiments.





**Figure 4** Comparison of light inhibition rate between ambient and VACES particles based on (a) hourly and (b) 30-minutes sampling.



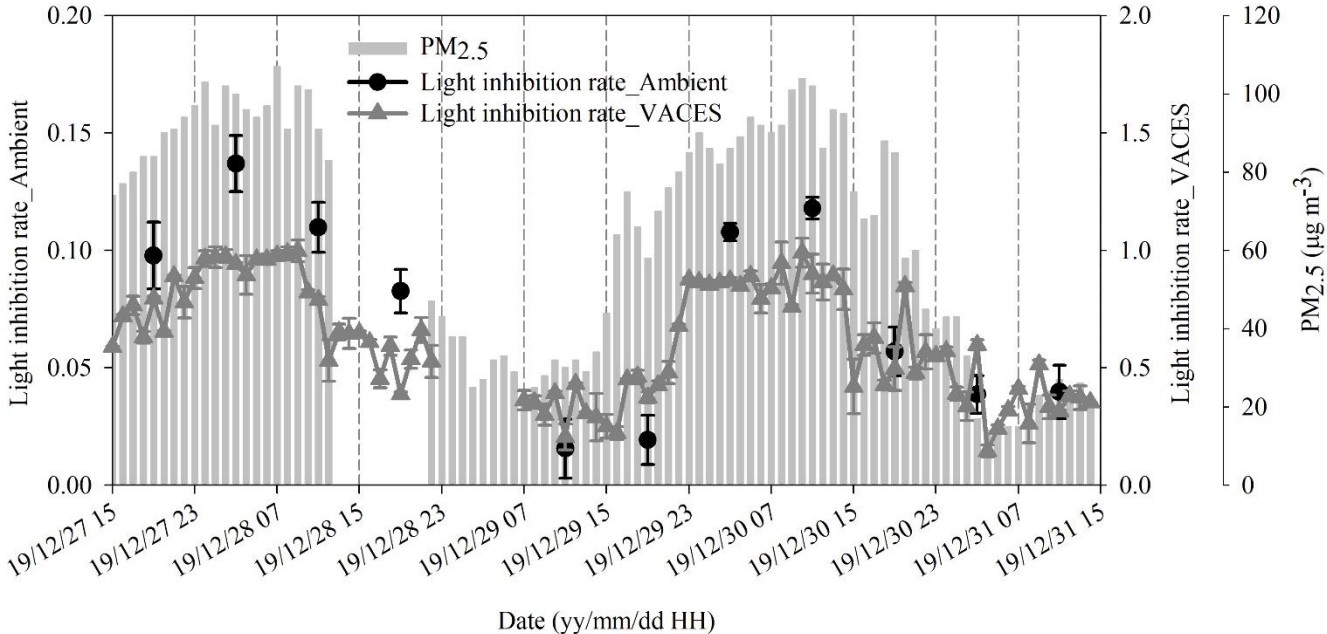

**Figure 5** Comparison of light inhibition rate between ambient and VACES particles based on continuous sampling of VACES

hourly and ambient 8 hours.





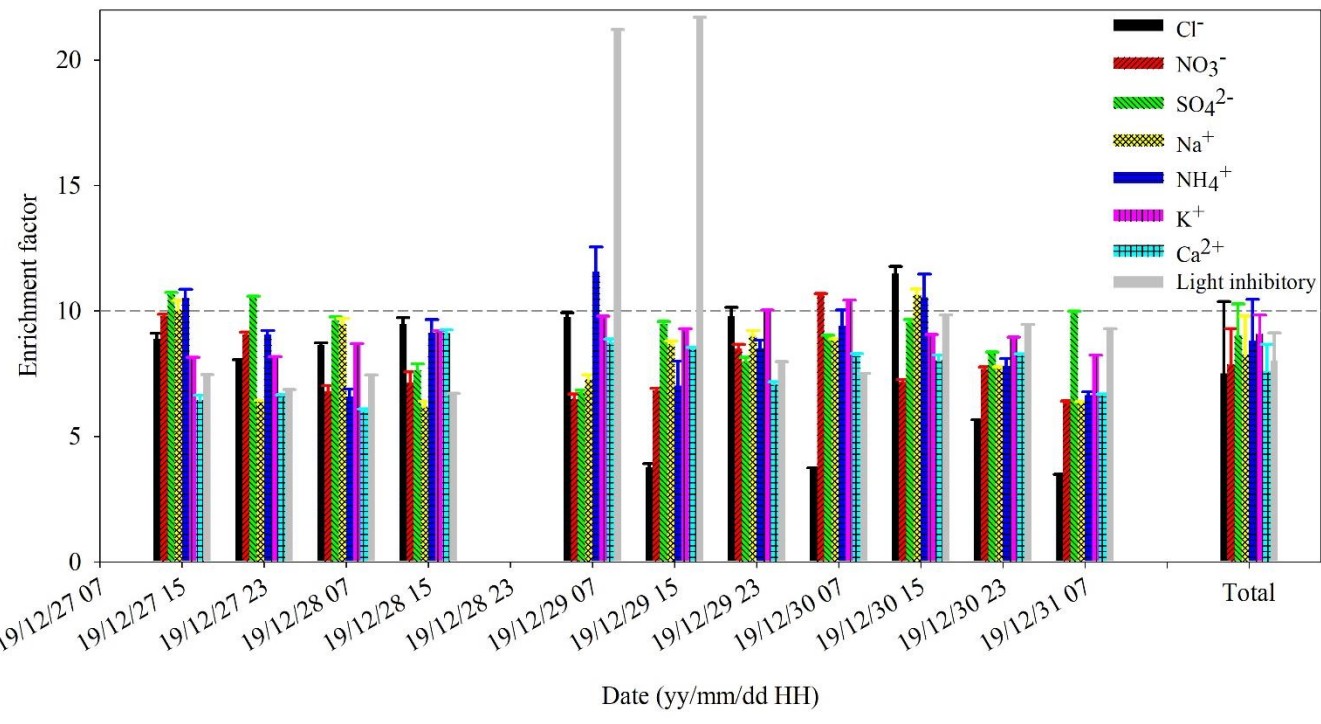


**Figure 6** Enrichment factors of chemical compositions and light inhibitory of $PM_{2.5}$ during continuous sampling period.