# Peer review of "A semi-continuous study on the ecotoxicity of atmospheric particles using a versatile aerosol concentration enrichment system (VACES): development and field characterization"

_Atmospheric Measurement Techniques, 2020_

## Referee Comment (RC1) · Anonymous Referee #1 · 13 May 2020

General comments : This work investigated a semi-continous system intending to online monitor the toxicity of atmospheric particles. An existing device, the VACES (versatile aerosol concentration enrichement system) was extended and enhanced for this purpose. This study provided a comparison between the measurement of the "toxicity" through using concentrated aerosols in the VACES system and non-concentrated from ambient air.

The manuscript is generally well-written and is straightforward (maybe too much). Within the framework of analyses presented in this work, the results appear to be

sound. However, there are a number of major issues, and among them, the scientific choice of the toxicicological assay. These should be considered and addressed before the manuscript can be considered for publication.

Specific concerns :

A major issue is that throughout the manuscript, the authors emphazied that PM health effects, may be measured by photobacteria assay and to my knowledge, this bioassay has not been shown in any study to be associated with adverse PM health effects (if this is incorrect, please provide relevant citations). The study relies on the assay developed by Jing et al, 2019 which is in fact an ecotoxicological assay, not a toxicological assay. This ecotoxicological assay is based on the light inhibition of photobacteria which is sensitive to most of environmental toxics. This assay is not specific and responds to many toxics when it's known that PM health effects rely mainly on oxidative stress. And finally this ecotoxicological assay (called biotoxic assay by their authors Jing et al : 2019) has not been compared to any toxicological assay. It does not guite make sense that the undertone of the manuscript hinges on the ability of VACES to permit monitoring aerosol toxicity when there is currently no link between photobacteria inhibition and adverse health effects from PM (if not relevant, please provide citatioon or comparison between photobacteria answer and toxicological results towards atmospheric particles) Another point is the comparison of the "ecotoxicity" between non-concentrated and concentrated aerosols in ambient air. During both experiments are done for a temperature of 45°±2 °C in the saturator and then results are compared for ambient samples and samples through VACES. I guess that this temperature is not physiologically relevant when aiming at monitoring human health. The system shouldn't overpass 37.5°C since at 45°C, many semi-volatile components may disappear and influence the answer of the system. Finally, the data presented are good, but the manuscript should be modified/re-written to emphasize on the measurements and data and not over-extrapolate the impacts and implications of the results to human health. All the more, analysis and VACES performances should be deeper.

---

## Author Comment (AC1) · 12 Jun 2020

**Correspondence to Review 1**

Thank you very much for your thorough and constructive comments on our manuscript amt-2020-10, entitled "A semi-continuous study on the toxicity of atmospheric particles using a versatile aerosol concentration enrichment system (VACES): development and field characterization". We made all corrections and revised the manuscript according to your comments. The response is given to each comment. In the revised manuscript, changes including some technical corrections are colored in blue.

**General comments:** This work investigated a semi-continuous system intending to online monitor the toxicity of atmospheric particles. An existing device, the VACES (versatile aerosol concentration enrichment system) was extended and enhanced for this purpose. This study provided a comparison between the measurement of the "toxicity" through using concentrated aerosols in the VACES system and non-concentrated from ambient air. The manuscript is generally well-written and is straightforward (maybe too much). Within the framework of analyses presented in this work, the results appear to be sound. However, there are a number of major issues, and among them, the scientific choice of the toxicicological assay. These should be considered and addressed before the manuscript can be considered for publication.

**Response:** As you pointed out, we give a specific description of the scientific choice of the toxicological assay (photobacteria) in the following responses and also add relevant explanations in manuscript.

**Specific concerns:**

**Comment 1:** A major issue is that throughout the manuscript, the authors emphasized that PM health effects, may be measured by photobacteria assay and to my knowledge, this bioassay has not been shown in any study to be associated with adverse PM health effects (if this is incorrect, please provide relevant citations). The study relies on the assay developed by Jing et al, 2019 which is in fact an ecotoxicological assay, not a toxicological assay. This ecotoxicological assay is based on the light inhibition of photobacteria which is sensitive to most of environmental toxics. This assay is not

specific and responds to many toxics when it's known that PM health effects rely mainly on oxidative stress. And finally this ecotoxicological assay (called biotoxic assay by their authors Jing et al., 2019) has not been compared to any toxicological assay. It does not quite make sense that the undertone of the manuscript hinges on the ability of VACES to permit monitoring aerosol toxicity when there is currently no link between photobacteria inhibition and adverse health effects from PM (if not relevant, please provide citation or comparison between photobacteria answer and toxicological results towards atmospheric particles)

**Response 1:** According to your questions mentioned in Comment 1, we give answers from four points:

First, it remains a scientific issue in vitro experiments that there is a lack of direct data support of the relationship between toxicity (e.g., cytotoxicity and ecotoxicity) and adverse PM health effects. Even for the exposure experiments (e.g., fish and mammalian), to our knowledge, no study exposes animals and human simultaneously to PMs. In fact, the definition of health effects are changes in health resulting from exposure to a source, which does not specifically refer to humans. However, in toxicology research, once health effects are mentioned, many people first think of human health effects. Therefore, in the absence of studies on the correlation between ecotoxicity and human health effects, we adopted your comments to remove or modify all discussions on health effects, and focus on the existing ecotoxicity results of PMs in order to avoid from misunderstanding.

Second is the use of the "toxicity" in the manuscript. It's absolutely right that the photobacteria experiment is an ecotoxicological assay, therefore, we change all "toxicity" words to "ecotoxicity" in manuscript as typically used in previous studies.

Third, there are several studies reported significant correlations between the Microtox (Photobacterium phosphoreum) $EC_{50}$ and rat/mouse $LD_{50}$ values (e.g., Fort, 1992; Kaiser et al., 1994), indicating the feasibility of photobacteria-base method on evaluating toxicity (exactly ecotoxicity).

Finally, the method of measuring ecotoxicity using photobacteria has long been routinely applied for water and soil research. This method has been standardized by the International Standards Organization (ISO 21338:2010: Water quality - Kinetic determination of the inhibitory effects of sediment, other solids and colored samples on the light emission of *Vibrio Fischeri*/ kinetic luminescent bacteria test. Photobacteria are also often used to assess the ecotoxicity of particulate matter and chemical components in ambient air. For instance, Turóczi et al. (2012) used *Vibrio fishcer* to

study the ecotoxicity of $PM_{10}$. This study directly evaluated the overall ecotoxicity of particles from different sources and seasons. Tositti et al. (2018) developed an ecotoxicity detection method using *Vibrio fishcer*, and found that ecotoxicity was closely related to the compositions of $PM_{10}$. Wang et al. (2016) demonstrated that the $PM_{2.5}$ components analyzed by *Photobacterium Phosphoreum T3* bioassay is ecologically toxic. Eck-Varanka et al. (2019) analyze the ecotoxicity of size-fractionated particles using *Vibrio fischeri*. Such literatures prove the feasibility of the photobacteria-based method in assessing the ecological toxicity of atmospheric particulate matter. The relevant description of the ecotoxicity assay of PMs in previous studies were also summarized and added in manuscript (Introduction section).

References

Eck-Varanka, B., Hubai, K., Horváth, E., Kováts, N., Teke, G., and Tóth, Á.: Assessing Ecotoxicity of Size-fractionated Airborne Particulate Matter, E3S Web Conf., 99, 2019.
Fort, F.: Correlation of Microtox EC, with mouse LD, Toxicol. In Vitro, 5, 73–82, 1992.
Kaiser, K. L., McKinnon, M. B., and Fort, F. L.: Interspecies toxicity correlations of rat, mouse and Photobacterium phosphoreum, Environ. Toxicol. Chem., 13(10), 1599-1606, 1994.
Tositti, L., Brattich, E., Parmeggiani, S., Bolelli, L., Ferri, E., and Girotti, S.: Airborne particulate matter biotoxicity estimated by chemometric analysis on bacterial luminescence data, Sci. Total Environ., 640, 1512-1520, 2018.
Turóczi, B., Hoffer, A., Tóth, Á., Kováts, N., Ács, A., Ferincz, Á., Kovács, A., and Gelencsér, A.: Comparative assessment of ecotoxicity of urban aerosol, Atmos. Chem. Phys., 12, 7365–7370, 2012.
Wang, W., Shi, C., Yan, Y., Yang, Y., and Zhou, B.: Eco-toxicological bioassay of atmospheric fine particulate matter (PM2.5) with Photobacterium Phosphoreum T3, Ecotox. Environ. Safe., 133, 226-234, 2016.

**Comment 2:** Another point is the comparison of the "ecotoxicity" between non-concentrated and concentrated aerosols in ambient air. During both experiments are done for a temperature of 45◦±2 ◦C in the saturator and then results are compared for ambient samples and samples through VACES. I guess that this temperature is not physiologically relevant when aiming at monitoring human health. The system shouldn't overpass 37.5◦C since at 45◦C, many semi-volatile components may disappear and influence the answer of the system.

**Response 2:** Yes, you are absolutely correct. A current publication showed that as the temperature increased to 50 °C, the concentration of particle number, mass, semi-VOCs, and volatile ions in the VACES system was lost by 50% (Pirhadi et al., 2020). However, in our study:

On one hand, the water in saturator was heated to 45±2 °C, but after mixing with ambient aerosol stream (ambient temperature was no more than 24 °C during experiment period), the temperature of the saturator decreased to 31 °C or lower. In Dameto de España's study, they emphasized that "this saturator temperature difference has a strong influence on the outlet temperature at the exit of the condenser and consequently on the actual supersaturation experienced by the particles". In our research, the temperature difference was also found, and the actual supersaturation temperature (vapor temperature) of the ambient particles was only 31 °C (after mixing with the ambient air). Since the temperature of the aerosol stream at the outlet of the saturator cannot be controlled, the only temperature that can be controlled is the water temperature, which should not only ensure the enrichment factor of the particulate concentration (~ tenfold), but also maintain a temperature similar to the ambient temperature (just as you mentioned, when conducting human health studies, the temperature should be below 37.5 °C). Therefore, even when studying the effects of atmospheric particles on human health, toxicity or ecotoxicity, our setting of 45±2 °C (the actual supersaturation temperature is 31 °C) is reasonable. Moreover, at the current temperature, the enrichment efficiency of PM in VACES reached 75-98% (size-dependent) as shown in Table 1, which was comparable with those studies setting temperature at ~ 35 °C.

On the other hand, the ambient filter samples were sonicated in an ultrasonic bath set at 45°C. In order to fix the bottle (with filters) in water and maintain a temperature similar to the supersaturation temperature in VACES, we put the bottle in a plastic box, the temperature of the water in the box did not exceed 34 °C.

The above details were not explained in the manuscript, which caused readers to misunderstand. Therefore, we added the above descriptions to the manuscript (Section 2.2 and 3.1).

References

Pirhadi, M., Mousavi, A., Taghvaee, S., Shafer, M. M., and Sioutas, C.: Semi-volatile components of PM2.5 in an urban environment: Volatility profiles and associated

oxidative potential, Atmos. Environ., 223, 117197, 2020.

Dameto de España, C., Steiner, G., Schuh, H., Sioutas, C., and Hitzenberger, R.: Versatile aerosol concentration enrichment system (VACES) operating as a cloud condensation nuclei (CCN) concentrator: development and laboratory characterization, Atmos. Meas. Tech., 12, 4733-4744, 2019.

**Comment 3:** Finally, the data presented are good, but the manuscript should be modified/re-written to emphasize on the measurements and data and not over-extrapolate the impacts and implications of the results to human health.

**Response 3:** We completely agree with your opinion. As you figured out in Comment 1, we have no further data to explain the relationship between ecotoxicity and health effect (It's a work of our follow-up research), thereby we emphasize on the existing data and delete/re-write all health-related descriptions as given in response 1.

**Comment 4:** All the more, analysis and VACES performances should be deeper.

**Response 4:** As a technical paper, we aimed to simply provide the technical parameters of the VACES-ecotoxicity system and clarify its feasibility in preliminary laboratory and field measurements. The feasibility of the system was mainly clarified from two aspects: 1) The system steadily concentrated the concentration of PM to nearly 10 times; 2) Even if the concentration of PM was very low (a common case in the atmosphere), PM via concentration can meet the detection limit of ecotoxicity. These two points are very important for further online measurement of PM ecotoxicity in ambient air. In this study, we obtained a stable enrichment effect, and when the environmental PM concentration was below the detection limit, the ecological toxicity of PM (0.5 h or 1 h sampling) was much higher than the detection limit. Then, the feasibility of the system was verified.

These summary descriptions were added to the manuscript (Conclusion section) to emphasize the key points and make the system performance analysis more in-depth and clearer. In addition, as you suggested, in future research, we will conduct a more in-depth analysis, focusing on the application of VACES-ecotoxic system in atmosphere and combining with other factors (such as chemical composition, gaseous precursors and meteorological parameters) to investigate the characteristics of PM ecotoxicity (semi-continuous or even online).

---

## Referee Comment (RC2) · Anonymous Referee #2 · 4 Dec 2020

The manuscript represents the results obtained by using a versatile aerosol concentration enrichment system (VACES), which was extended in order to estimate the toxicity of ambient air. The differences between concentrated and non-concentrated particulate matter were manifested in the measured aerosol properties; while offline and online sampling equipment has been applied to investigate in detail the acquired toxicity data. The observed results were further enhanced by studying the correlations between the measured toxicity and all obtained physical/chemical aerosol parameters, in order to highlight the role of aerosol enrichment through the VACES technique. Overall,

the results are interesting and well investigated but the manuscript is difficult to follow. An important vantage of this study is that it includes a variety of techniques, providing several aerosol properties that may capture the effect of VACES on sampled particulate matter. Finally, it is essential that this study attempts to highlight the role of aerosol chemical components on the measured toxicity levels. Nevertheless, the manuscript needs some major revision regarding the selected toxicity assay, the fact that the utilized techniques are not clearly described and, in several places the justifications or discussions should be presented in a more comprehensive way.

Some indicative major issues to be addressed are the following: Section 1: Please justify the application of photobacteria in atmospheric particle toxicity and its association with health effects. Section 2.2: It is difficult to follow the description and the setup of the instrumentation. Please rephrase to clarify the definition of the optimal technical parameters, the calculation of measured theoretical concentrations, the utility of ambient aerosol removal through filter, the description of performance testing, the essential technical difference between offline and online collection, the justification of selected sampling parameters (both offline and online). Please rephrase to describe the obtained samples since there are collections mentioned as continuous, based on 1min, on 8hrs, on 1hr and on 30min. Section 2.3: As already mentioned in Section 1, the selected toxicity assay is mentioned in Jang et al., (2019) as a biotoxic assay based on a luminescent bacterium. Please justify the selection of this particular assay and clarify how this could be linked to health effects. Please mention whether this assay has been part of epidemiological studies or has been compared / combined with any cellular or acellular aerosol toxicity assay linked to health effects. Section 3.3: The results should be clearly described and justified in a comprehensive way. For example, please rephrase to clarify clean and polluted days, meaning of toxicity of non-ambient samples, normal fluctuation range of the luminescence bacterium affecting the sample uncertainty, effect of low toxicity of ambient samples on concentration, differences in variability between EFs and ambient and VACES samples, effect of enrichment in chemical composition of samples. Please note that it would be helpful to add a more

detailed description in the legends of the figures, regarding the properties of the presented samples.

---

## Author Comment (AC2) · 12 Dec 2020

Dear Reviewer1,

Based on Reviewer2's comments, we once again made major revisions to previous responses and manuscripts. Attached is the updated response (revisions are marked in blue).

Thanks for your second comment!

[Figure]

Best regards,

Xiaona Shang

Please also note the supplement to this comment:
https://amt.copernicus.org/preprints/amt-2020-10/amt-2020-10-AC2-supplement.pdf

––––––––––––––––––––––––

**Supplement:**

**Correspondence to Review 1**

Thank you very much for your thorough and constructive comments on our manuscript amt-2020-10, entitled "A semi-continuous study on the toxicity of atmospheric particles using a versatile aerosol concentration enrichment system (VACES): development and field characterization". We made all corrections and revised the manuscript according to your comments. The response is given to each comment. In the revised manuscript, changes including some technical corrections are colored in blue.

**(Note: Re-added or modified responses have been marked in blue below.)**

**General comments:** This work investigated a semi-continuous system intending to online monitor the toxicity of atmospheric particles. An existing device, the VACES (versatile aerosol concentration enrichment system) was extended and enhanced for this purpose. This study provided a comparison between the measurement of the "toxicity" through using concentrated aerosols in the VACES system and non-concentrated from ambient air. The manuscript is generally well-written and is straightforward (maybe too much). Within the framework of analyses presented in this work, the results appear to be sound. However, there are a number of major issues, and among them, the scientific choice of the toxicicological assay. These should be considered and addressed before the manuscript can be considered for publication.

**Response:** As you pointed out, we gave a specific description of the scientific choice of the toxicological assay (photobacteria) in the following responses and also added relevant explanations in manuscript.

**Specific concerns:**

**Comment 1:** A major issue is that throughout the manuscript, the authors emphasized that PM health effects, may be measured by photobacteria assay and to my knowledge, this bioassay has not been shown in any study to be associated with adverse PM health effects (if this is incorrect, please provide relevant citations). The study relies on the assay developed by Jing et al, 2019 which is in fact an ecotoxicological assay, not a toxicological assay. This ecotoxicological assay is based on the light inhibition of photobacteria which is sensitive to most of environmental toxics. This assay is not specific and responds to many toxics when it's known that PM health effects rely mainly on oxidative stress. And finally this ecotoxicological assay (called biotoxic assay by their authors Jing et al., 2019) has not been compared to any toxicological assay. It does not quite make sense that the undertone of the manuscript hinges on the ability of VACES to permit monitoring aerosol toxicity when there is currently no link between photobacteria inhibition and adverse health effects from PM (if not relevant, please provide citation or comparison between photobacteria answer and toxicological results towards atmospheric particles)

**Response 1:** According to your questions mentioned in Comment 1, we gave answers from four points:

First, it remains a scientific issue in vitro experiments that there was a lack of direct data support of the

relationship between toxicity (e.g., cytotoxicity and ecotoxicity) and adverse PM health effects. Even for the exposure experiments (e.g., fish and mammalian), to our knowledge, no study exposed animals and human simultaneously to PMs due to ethics. In fact, the definition of health effects are changes in health resulting from exposure to a source, which does not specifically refer to humans. However, in toxicology research, once health effects are mentioned, many people first think of human health effects. Therefore, in the absence of studies on the correlation between ecotoxicity and human health effects, we adopted your comments to remove or modify all discussions on health effects, and focused on the existing ecotoxicity results of PMs in order to avoid from misunderstanding.

Second is the use of the "toxicity" in the manuscript. It's absolutely right that the photobacteria experiment is an eco-toxicological assay, therefore, we changed all "toxicity" words to "ecotoxicity" in manuscript as typically used in previous studies.

Third, there are several studies reported significant correlations between the Microtox (Photobacterium phosphoreum) $EC_{50}$ and rat/mouse $LD_{50}$ values (e.g., Fort, 1992; Kaiser et al., 1994), indicating the feasibility of photobacteria-base method on evaluating toxicity (exactly ecotoxicity).

Finally, the method of measuring ecotoxicity using photobacteria has long been routinely applied for water and soil research. This method has been standardized by the International Standards Organization (ISO 21338:2010: Water quality - Kinetic determination of the inhibitory effects of sediment, other solids and colored samples on the light emission of *Vibrio Fischeri*/ kinetic luminescent bacteria test. Photobacteria were also often used to assess the ecotoxicity of particulate matter and chemical components in ambient air. For instance, Turóczi et al. (2012) used *Vibrio fishcer* to study the ecotoxicity of $PM_{10}$. This study directly evaluated the overall ecotoxicity of particles from different sources and seasons. Tositti et al. (2018) developed an ecotoxicity detection method using *Vibrio fishcer*, and found that ecotoxicity was closely related to the compositions of $PM_{10}$. Wang et al. (2016) demonstrated that the $PM_{2.5}$ components analyzed by *Photobacterium Phosphoreum T3* bioassay is ecologically toxic. Eck-Varanka et al. (2019) analyzed the ecotoxicity of size-fractionated particles using *Vibrio fischeri*. Such literature proved the feasibility of the photobacteria-based method in assessing the ecological toxicity of atmospheric particulate matter. The relevant descriptions of the ecotoxicity assay of PMs in previous studies were also summarized and added in manuscript (Introduction section).

All the replies mentioned above were already in the "Introduction" section or have been added. The blue font below shows better arrangements and revisions to the introduction:

To solve this problem, photobacteria (e.g. Photobacterium phosphoreum) are utilized in the ecotoxicity study of atmospheric particles, because the detection was rapid (e.g. within 15 minutes; Hoover et al., 2005) and the cultivation period is only 5 minutes (Jing et al., 2019). The method of measuring ecotoxicity using photobacteria bioluminescence inhibition bioassay has long been routinely applied and standardized for water and soil research (ISO 21338:2010: Water quality – Kinetic determination of the inhibitory effects of sediment, other solids and coloured samples on the light emission of Vibrio fischeri /kinetic luminescent bacteria test/). It had been reported that the photobacterium phosphoreum $EC_{50}$ (median effective concentration) significantly correlated to rat and mouse $LD_{50}$ (the lethal dose for 50 percent of the animals tested) values, indicating the reliability of photobacteria-based ecotoxicity assay (Fort, 1992; Kaiser et al., 1994). Recently, photobacteria have also been often used to assess the ecotoxicity of particulate matter and chemical components in atmosphere. For instance, Turóczi et al. (2012) used *Vibrio fishcer* to study the ecotoxicity of $PM_{10}$. This study directly evaluated the overall ecotoxicity of particles from different sources and seasons. Tositti et al. (2018) developed an ecotoxicity detection method using *Vibrio fishcer*, and found that ecotoxicity was closely related to the compositions of $PM_{10}$. Wang et al. (2016) demonstrated that the $PM_{2.5}$ components analyzed by *Photobacterium Phosphoreum T3* bioassay was ecologically toxic. Eck-Varanka et al. (2019) analyze the ecotoxicity of size-fractionated particles using *Vibrio fischeri*. Such literature proved the feasibility of the photobacteria-based method in assessing the ecological toxicity of atmospheric particulate matter. However, the detection limit of ecotoxicity using photobacteria is high. For example, in Jing's research, samples with a light inhibitory rate of less than 20 % were considered to be non-toxic due to the impact of normal bacteria fluctuations. Whereas, the concentration of atmospheric aerosols is usually far lower than that required for eco-toxic assay in case of short sampling time (e.g. one hour), which means a longer sampling time is required. Nevertheless, long-time sampling may lead to a large loss of volatile substances or chemical reactions in the particles, subsequently resulting in large errors in ecotoxicity analysis (Weiden et al., 2009).

**Comment 2:** Another point is the comparison of the "ecotoxicity" between non-concentrated and concentrated aerosols in ambient air. During both experiments are done for a temperature of 45∘±2 ∘C in the saturator and then results are compared for ambient samples and samples through VACES. I guess that this temperature is not physiologically relevant when aiming at monitoring human health. The system shouldn't overpass 37.5∘C since at 45∘C, many semi-volatile components may disappear and influence the answer of the system.

**Response 2:** Yes, you are absolutely correct. A current publication showed that as the temperature increased to 50 °C, the concentration of particle number, mass, semi-VOCs, and volatile ions in the VACES system was lost by 50% (Pirhadi et al., 2020). However, in our study:

On one hand, the water in saturator was heated to 45±2 °C, but after mixing with ambient aerosol stream (ambient temperature was no more than 24 °C during experiment period), the temperature of the saturator decreased to 31 °C or lower. In Dameto de España's study, they emphasized that "this saturator

temperature difference has a strong influence on the outlet temperature at the exit of the condenser and consequently on the actual supersaturation experienced by the particles". In our research, the temperature difference was also found, and the actual supersaturation temperature (vapor temperature) of the ambient particles was only 31 °C (after mixing with the ambient air). Since the temperature of the aerosol stream at the outlet of the saturator cannot be controlled, the only temperature that can be controlled was the water temperature, which not only ensured the enrichment factor of the particulate concentration (~ tenfold), but also maintain a temperature similar to the ambient temperature (just as you mentioned, when conducting human health studies, the temperature should be below 37.5 °C). Therefore, even when study the effects of atmospheric particles on human health, toxicity or ecotoxicity, our setting of 45±2 °C (the supersaturation temperature was 31 °C) is reasonable. Moreover, at the current temperature, the enrichment efficiency of PM in VACES reached 75-98% (size-dependent) as shown in Table 1, which was comparable with those studies setting temperature at ~ 35 °C.

On the other hand, the ambient filter samples were sonicated in an ultrasonic bath set at 45°C. In order to fix the bottle (with filters) in water and maintain a temperature similar to the supersaturation temperature in VACES, we put the bottle in a plastic box, the temperature of the water in the box did not exceed 34 °C.

The above details were not explained in the manuscript, which caused readers to misunderstand. Therefore, we added the above descriptions to the manuscript (Section 2.2 and 3.1).

**Comment 3:** Finally, the data presented are good, but the manuscript should be modified/re-written to emphasize on the measurements and data and not over-extrapolate the impacts and implications of the results to human health.

**Response 3:** We completely agree with your opinion. As you figured out in Comment 1, we have no further data to explain the relationship between ecotoxicity and health effect (It's a work of our follow-up research), thereby we emphasized on the existing data and delete/re-write all health-related descriptions as given in response 1. The deeper discussion of the current data was shown in the next response.

**Comment 4:** All the more, analysis and VACES performances should be deeper.

**Response 4:** The deeper (in blue) discussion focusing on the current data was added in Discussion and Conclusion section as follows,

[revised manuscript text omitted]

---

## Author Comment (AC3) · 12 Dec 2020

**Correspondence to Review 2**

Thank you very much for your thorough and constructive comments on our manuscript amt-2020-10, entitled "A semi-continuous study on the toxicity of atmospheric particles using a versatile aerosol concentration enrichment system (VACES): development and field characterization". We made all corrections and revised the manuscript according to your comments. The response is given to each comment. In the revised manuscript, changes including some technical corrections are colored in blue.

**General comments:** The manuscript represents the results obtained by using a versatile aerosol concentration enrichment system (VACES), which was extended in order to estimate the toxicity of ambient air. The differences between concentrated and non-concentrated particulate matter were manifested in the measured aerosol properties; while offline and online sampling equipment has been applied to investigate in detail the acquired toxicity data. The observed results were further enhanced by studying the correlations between the measured toxicity and all obtained physical/chemical aerosol parameters, in order to highlight the role of aerosol enrichment through the VACES technique. Overall, the results are interesting and well investigated but the manuscript is difficult to follow. An important vantage of this study is that it includes a variety of techniques, providing several aerosol properties that may capture the effect of VACES on sampled particulate matter. Finally, it is essential that this study attempts to highlight the role of aerosol chemical components on the measured toxicity levels. Nevertheless, the manuscript needs some major revision regarding the selected toxicity assay, the fact that the utilized techniques are not clearly described and, in several places the justifications or discussions should be presented in a more comprehensive way.

**Response:** Thank you for your recognition of our results and overall research work. In response to your professional opinions, we have made major revisions to the deficiencies in the original manuscript, including the detailed explanation of toxicity assay, more specific description of experimental techniques, rephrasing the justifications and discussions so that it is clear.

**Major issues:**

**Comment 1:** Section 1: Please justify the application of photobacteria in atmospheric particle toxicity and its association with health effects.

**Response 1:** Comment 1 is an important question, we reply as follows,

First is the application of photobacteria in atmospheric particle toxicity (exactly ecotoxicity, we changed toxicity to ecotoxicity in response to Reviewer 1). In fact, photobacteria have long been used in the

study of particle ecotoxicity and even related to specific chemical components in ambient air. Take the researches in recent years as an example, Turóczi et al. (2012) used *Vibrio fishcer* to study the ecotoxicity of $PM_{10}$. This study directly evaluated the overall ecotoxicity of particles from different sources and seasons. Tositti et al. (2018) developed an ecotoxicity detection method using *Vibrio fishcer*, and found that ecotoxicity was closely related to the compositions of $PM_{10}$. Wang et al. (2016) demonstrated that the $PM_{2.5}$ components analyzed by *Photobacterium Phosphoreum T3* bioassay was ecologically toxic. Eck-Varanka et al. (2019) analyzed the ecotoxicity of size-fractionated particles using *Vibrio fischeri*. Such literature proved the feasibility of the photobacteria-based method in assessing the ecological toxicity of atmospheric particulate matter. The relevant description of the ecotoxicity assay of PMs in previous studies were summarized and added in manuscript (Introduction section).

Second, it remains a scientific issue in vitro experiments that there is a lack of direct data support of the relationship between toxicity (e.g., cytotoxicity and ecotoxicity) and adverse PM health effects. Even for the exposure experiments (e.g., fish and mammalian), to our knowledge, no study exposes animals and human simultaneously to PMs due to ethics. Therefore, we removed all contexts on health effects and emphasized on the measurements and data to avoid from over-extrapolating the impacts and implications of the results to human health (as Reviewer 1 mentioned). Even so, the method of using photobacteira bacteria to study ecotoxicity or even cytotoxicity is feasible because: 1) the method of measuring ecotoxicity using photobacteria has long been routinely applied for water and soil research. This method has been standardized by the International Standards Organization (ISO 21338:2010: Water quality - Kinetic determination of the inhibitory effects of sediment, other solids and colored samples on the light emission of *Vibrio Fischeri*/ kinetic luminescent bacteria test; 2) many applications in previous studies on particle ecotoxixity as above mentioned; 3) there are also several studies reported strong correlations between the Microtox (Photobacterium phosphoreum) $EC_{50}$ and rat/mouse $LD_{50}$ values (e.g., Fort, 1992; Kaiser et al., 1994). The information was also summarized in manuscript (Introduction section)

References

Eck-Varanka, B., Hubai, K., Horváth, E., Kováts, N., Teke, G., and Tóth, Á.: Assessing Ecotoxicity of Size-fractionated Airborne Particulate Matter, E3S Web Conf., 99, 2019.

Fort, F.: Correlation of Microtox EC, with mouse LD, Toxicol. In Vitro, 5, 73–82, 1992.

Kaiser, K. L., McKinnon, M. B., and Fort, F. L.: Interspecies toxicity correlations of rat, mouse and Photobacterium phosphoreum, Environ. Toxicol. Chem., 13(10), 1599-1606, 1994.

Tositti, L., Brattich, E., Parmeggiani, S., Bolelli, L., Ferri, E., and Girotti, S.: Airborne particulate matter biotoxicity estimated by chemometric analysis on bacterial luminescence data, Sci. Total Environ., 640, 1512-1520, 2018.

Turóczi, B., Hoffer, A., Tóth, Á., Kováts, N., Ács, A., Ferincz, Á., Kovács, A., and Gelencsér, A.: Comparative assessment of ecotoxicity of urban aerosol, Atmos. Chem. Phys., 12, 7365–7370, 2012.

Wang, W., Shi, C., Yan, Y., Yang, Y., and Zhou, B.: Eco-toxicological bioassay of atmospheric fine particulate matter (PM2.5) with Photobacterium Phosphoreum T3, Ecotox. Environ. Safe., 133, 226-234, 2016.

**Comment 2:** Section 2.2: It is difficult to follow the description and the setup of the instrumentation. Please rephrase to clarify the definition of the optimal technical parameters, the calculation of measured theoretical concentrations, the utility of ambient aerosol removal through filter, the description of performance testing, the essential technical difference between offline and online collection, the justification of selected sampling parameters (both offline and online). Please rephrase to describe the obtained samples since there are collections mentioned as continuous, based on 1min, on 8hrs, on 1hr and on 30min.

**Response 2:** As pointed out in Comment 2, we rephrased Methodology, and all above mentioned points in bold were explained in details and in a better arrangement. Moreover, in order to look more intuitive and clearer and easier to explain, we modified the set-up structure in Figure 1 and added necessary legends and a full explanation in the caption of the figure.

The revised Methodology in blue:

[revised manuscript text omitted]

160 **Comment 3:** Section 2.3: As already mentioned in Section 1, the selected toxicity assay is mentioned in Jang et al., (2019) as a biotoxic assay based on a luminescent bacterium. Please justify the selection of this particular assay and clarify how this could be linked to health effects. Please mention whether this assay has been part of epidemiological studies or has been compared / combined with any cellular or acellular aerosol toxicity assay linked to health effects.

165 **Response 3:** For the first question, the reason why we selected the biotoxicity test based on luminescent bacteria is that the detection method is rapid and the luminescent bacteria are sensitive to changes in the concentration of toxic components in particles. The luminescent bacteria assay is an acute toxicity detection method. Its maximum exposure time is 24 hours, and its adverse effect (bacterial death) should occur within 14 days. The ecotoxicity determination in this study could be completed within 15 170 minutes, while the culture time of the bacteria was only 5 minutes. In addition, Figure 5 in the original manuscript showed that changes in ecotoxicity are very sensitive to changes in $PM_{2.5}$. In order to achieve online ecotoxicity measurement, we decided to use this method initially.

Regarding the second question, this is a toxicology study, which is not directly related to health effects in this study or previous studies revised in response 1. Therefore, in order to focus on the results and 175 avoid exaggerating the significance of this study, we have deleted all discussions about health effects.

Regarding the third question, there is still a lack of research reports based on the direct relationship between the ecotoxicity of bacteria and the health effects of atmospheric particles. Because low concentrations of atmospheric particulate matter require a long exposure time to have health effects, but bacteria are very sensitive and their exposure time is less than 24 hours (that is, the big difference in exposure time will make it difficult to conduct correlation studies between the two). However, several studies have reported a close correlation between Microtox (phosphobacteria) EC50 and rat/mouse LD50 values (for example, Fort, 1992; Kaiser et al., 1994), as described in response 1.

All the replies mentioned above are already in the "Introduction" section or have been added. The blue font below shows better arrangements and revisions to the introduction:

**1 Introduction**

Currently, most toxicological studies focus on discovering the relationship between particulate matter and the morbidity or mortality of organisms (e.g. Vincent et al., 2001; Cox et al., 2016; Miri et al., 2018), or exploring toxic mechanisms by exposure experiments (e.g. Magnani et al., 2016; Huang et al., 2017; Rychlik et al., 2019). However, the measurement ecotoxicity data are rarely available because of technical limitations. For instance, it requires a long detection time due to the animal and plant reproduction or cell cultivation (National Research Council, 2006), but the concentration and chemical composition of particulate matter in the atmosphere continue to change over time, especially during severe pollution (Shang et al., 2018a, b). Thereby, a short analyzing time is quite important.

To solve this problem, photobacteria (e.g. Photobacterium phosphoreum) are utilized in the ecotoxicity study of atmospheric particles, because the detection was rapid (e.g. within 15 minutes; Hoover et al., 2005) and the cultivation period is only 5 minutes (Jing et al., 2019). The method of measuring ecotoxicity using photobacteria bioluminescence inhibition bioassay has long been routinely applied and standardized for water and soil research (ISO 21338:2010: Water quality – Kinetic determination of the inhibitory effects of sediment, other solids and coloured samples on the light emission of Vibrio fischeri /kinetic luminescent bacteria test/). It had been reported that the photobacterium phosphoreum $EC_{50}$ (median effective concentration) significantly correlated to rat and mouse $LD_{50}$ (the lethal dose for 50 percent of the animals tested) values, indicating the reliability of photobacteria-based ecotoxicity assay (Fort, 1992; Kaiser et al., 1994). Recently, photobacteria have also been often used to assess the ecotoxicity of particulate matter and chemical components in atmosphere. For instance, Turóczi et al. (2012) used *Vibrio fishcer* to study the ecotoxicity of $PM_{10}$. This study directly evaluated the overall ecotoxicity of particles from different sources and seasons. Tositti et al. (2018) developed an ecotoxicity detection method using *Vibrio fishcer*, and found that ecotoxicity was closely related to the compositions of $PM_{10}$. Wang et al. (2016) demonstrated that the $PM_{2.5}$ components analyzed by *Photobacterium Phosphoreum T3* bioassay was ecologically toxic. Eck-Varanka et al. (2019) analyze the ecotoxicity of size-fractionated particles using *Vibrio fischeri*. Such literature proved the feasibility of the photobacteria-based method in assessing the ecological toxicity of atmospheric particulate matter. However, the detection limit of ecotoxicity using photobacteria is high. For example, in Jing's research, samples with a light inhibitory rate of less than 20 % were considered to be non-toxic due to the impact of normal bacteria fluctuations. Whereas, the concentration of atmospheric aerosols is usually far lower than that required for eco-toxic assay in case of short sampling time (e.g. one hour), which means a longer sampling time is required. Nevertheless, long-time sampling may lead to a large loss of volatile substances or chemical reactions in the particles, subsequently resulting in large errors in ecotoxicity analysis (Weiden et al., 2009).

215    In this respect, aerosol enrichment techniques have been developed and applied to increase aerosol concentrations to meet ecotoxicity detection limits……

**Comment 4:** Section 3.3: The results should be clearly described and justified in a comprehensive way. For example, please rephrase to clarify clean and polluted days, meaning of toxicity of non-ambient
220    samples, normal fluctuation range of the luminescence bacterium affecting the sample uncertainty, effect of low toxicity of ambient samples on concentration, differences in variability between EFs and ambient and VACES samples, effect of enrichment in chemical composition of samples. Please note that it would be helpful to add a more detailed description in the legends of the figures, regarding the properties of the presented samples.

225    **Response 4:** According to EPA standards, $PM_{2.5}$ does not exceed 35 µg/m$^3$ in a 24-hour period. Therefore, our original classification standard is that the days less than or equal to 35 µg/m$^3$ are clean days, and the days greater than 35 µg/m$^3$ are polluted days. However, due to the limitation of the amount of data, original manuscript did not perform a divisional chemical or ecotoxicity analysis on clean and polluted days. Therefore, we deleted the information of pollution classification and only
230    retained the $PM_{2.5}$ concentration range to emphasize that even at low PM concentrations, this integrated detection method could also detect its ecotoxicity.

The "non-ambient samples" was changed to "VACES samples" which was defined in revised Methodology section (with ten times the concentration enrichment in VACES).

We changed the expression from "normal fluctuation range of the luminescence bacterium affecting the

[revised manuscript text omitted]

In addition, apart from the upper revision of Figure 1, we also added some legend explanation in figure captions like:

**Figure 4** Comparison of light inhibition rate and ratio of ambient and VACES particles with ambient PM$_{2.5}$ concentration based on (a) hourly and (b) 30 min discontinuous sample collection during 23$^{rd}$ October–11$^{st}$ December, 2019 in Shanghai, China. Baseline reflected the accuracy of photobacteria based ecotoxicity assay method and below the baseline, the accuracy is low.

**Figure 5** Comparison of light inhibition rate between ambient and VACES particles based on continuous sampling of VACES and ambient. VACES samples were collected hourly and ambient filter samples were collected every eight hours. The PM$_{2.5}$ concentration data was collected hourly from a nearby monitoring center (online data).

**Figure 6** Enrichment factors of chemical compositions and light inhibitory of PM$_{2.5}$ during continuous sampling period. The EF was calculated by the ratio of chemical concentrations of VACES to ambient particles. The component concentration of VACES particles was one hour per sample, and the concentration of ambient particles was 8 h per sample. For the ratio, we averaged the concentrations of VACES samples every 8 h to correspond to that of ambient samples.